# Influence of Burnout on Patient Safety: Systematic Review and Meta-Analysis

**DOI:** 10.3390/medicina55090553

**Published:** 2019-08-30

**Authors:** Cíntia de Lima Garcia, Luiz Carlos de Abreu, José Lucas Souza Ramos, Caroline Feitosa Dibai de Castro, Fabiana Rosa Neves Smiderle, Jaçamar Aldenora dos Santos, Italla Maria Pinheiro Bezerra

**Affiliations:** 1Laboratório de Delineamento de Estudos e Escrita Científica, Centro Universitário Saúde ABC (CUSABC) Convênio Acre/FMABC-007/2015, Santo André, São Paulo 09060-870, Brazil; 2Departamento de Enfermagem, Faculdade de Juazeiro do Norte (FJN), Juazeiro do Norte, Ceará 63010-475, Brazil; 3Departamento de Enfermagem. Faculdade de Medicina ESTACIO de Juazeiro do Norte (ESTACIO FMJ), Juazeiro do Norte, Ceará 63.048-080, Brazil; 4Programa de Mestrado em Políticas Públicas e Desenvolvimento Local, Escola Superior de Ciências da Santa Casa de Misericórdia (EMESCAM), Vitória, Espírito Santo 29045-402, Brazil; 5Laboratório de Escrita Científica, Escola Superior de Ciências da Santa Casa de Misericórdia (EMESCAM), Vitória, Espírito Santo 29045-402, Brazil; 6Graduate Entry Medical School, University of Limerick, V94 T9PX Limerick, Ireland; 7Departamento de Enfermagem. Universidade Federal do Acre (UFAC), Rio Branco, Acre 69.915-900, Brazil; 8Programa de Mestrado em Ciências da Saúde da Amazônia, Bolsista CAPES Brasil, Universidade Federal do Acre (UFAC), Rio Branco, Acre 69.915-900, Brazil

**Keywords:** Patient safety, burnout professional safety, organizational culture, safety management

## Abstract

*Background and Objectives:* Several factors can compromise patient safety, such as ineffective teamwork, failed organizational processes, and the physical and psychological overload of health professionals. Studies about associations between burn out and patient safety have shown different outcomes. *Objective:* To analyze the relationship between burnout and patient safety. *Materials and Methods:* A systematic review with a meta-analysis performed using PubMed and Web of Science databases during January 2018. Two searches were conducted with the following descriptors: (i) patient safety AND burnout professional safety AND organizational culture, and (ii) patient safety AND burnout professional safety AND safety management. *Results:* Twenty-one studies were analyzed, most of them demonstrating an association between the existence of burnout and the worsening of patient safety. High levels of burnout is more common among physicians and nurses, and it is associated with external factors such as: high workload, long journeys, and ineffective interpersonal relationships. Good patient safety practices are influenced by organized workflows that generate autonomy for health professionals. Through meta-analysis, we found a relationship between the development of burnout and patient safety actions with a probability of superiority of 66.4%. *Conclusion:* There is a relationship between high levels of burnout and worsening patient safety.

## 1. Introduction

Patient safety is one of the main challenges for quality of care, as adverse events are common in health services. In the United States, medical errors account for more than 250,000 deaths a year, making it the third leading cause of death [1]. A study conducted in Norway has identified that patients who die in hospitals experience seven times the rate of serious adverse events [2].

Patient safety is highlighted as one of the main challenges nowadays, as in many cases, patient care—in addition to system failures, poor organizational processes, and faulty management—also depends on health professionals. Changing habits requires a better training of these professionals based on the assumption that several studies indicate that failures in patient safety are also a consequence of human failures regarding communication, teamwork, and psychological health among health professionals [3,4].

Based on evidence, training would be on clinical practice guidelines, adverse events, technologies used in the service, better working conditions, continuous guidance on infection prevention methods, and offering better psychological and emotional support to health professionals [3,4].

Professional well-being, depression, anxiety, and burnout syndrome are determining factors that influence the care provided to patients, although they have different characteristics among them [5,6]. It is suggested that there is a proven relationship between poor well-being and moderate to high levels of burnout and poor patient safety resulting in assistance errors [7].

Burnout is a disorder that is directly related to workplace conditions due to the occupational stress in which health and education workers are constantly under, mainly associated with interaction with other people; as such, this condition is deeply studied in these workers. Regarding their usual risks, health professionals are exhausted during work hours, and therefore will not be able to perform an effective care practice and may cause harm to both patients and health service in general [8,9].

A study by Roteinstein et al. found a large variability (0% to 80.5%) of this disorder among physicians [10]. Among nurses, a prevalence of 30% was found in at least one of the three main domains (depersonalization, emotional exhaustion, and low professional achievement) of the disturbance [11].

Patient safety permeates through organizational, social, and individual factors that depend on physical and human resources. Burnout also depends on these characteristics, such as physical organizational factors and human interactions; therefore there is an intrinsic link between these [7].

Adverse events are defined as possible complications that may arise from patient care, emerging from errors that do not belong to the natural history of the disease. When these are due to errors, they are called avoidable adverse events [12].

Thus, because of the high prevalence of burnout among health workers due to the increasing number of adverse events [1,2,3,4,5,6,7,8,9,10], the following question arises: What is the relationship between burnout and patient safety?

The exposed problem justifies the need for studies to elucidate the associations between patient safety and burnout since to the relationship between them is frequently evidenced. The findings of this research will contribute toward a better understanding of this issue and provide visibility to the subject.

Thus, the objective of this study is to analyze the relationship between burnout and patient safety.

## 2. Materials and Methods

This systematic review and meta-analysis was performed according the Preferential Reports for Systematic Reviews and Meta-Analysis (PRISMA). The study was conducted in January 2018.

Following a standardized Population, Intervention, Comparison and Outcome (PICO) format, this study investigates the relationship between burnout syndrome (population) in the context of patient safety (intervention) by observing the influence that this relationship can exert on the assistance (comparison) and analyzing the results to understand the problem (outcome).

### 2.1. Search Strategy

The articles were searched through the following databases, which have a huge scope in the research topic: Pubmed (http://www.ncbi.nlm.nih.gov/pubmed) and Web of Science (https://webofknowledge.com/). In both databases, two refining searches were carried out regarding human studies. The first used the following combination of the descriptors: patient safety AND burnout professional safety AND organizational culture. The second search was performed with the following combination: patient safety AND burnout professional safety AND safety management.

The burnout definition was taken from the Maslach Burnout Inventory (MBI) form published in 1986, which is the main burnout assessment tool to date [13,14,15]. For patient safety, we used the term applied in the Hospital Survey on Patient Safety Culture (HSOPSC) instrument [16,17,18]. The rest of terms were used to refine the search regarding professional practice.

Initially, a period for filtering articles was not restricted, given the need to search all available articles.

### 2.2. Data Extraction and Quality Assessment

The selection process began by reading the titles and summaries of each article. Second, after excluding abstracts that were not adequate, the articles were read in full.

Data on method, participants, interventions, and outcomes were analyzed and extracted by two independent reviewers. In order to assess the risk of bias and the quality of the study, the Chochrane Manual for Systematic Intervention Reviews [19] was used, where any discrepancies were resolved by a third researcher.

### 2.3. Eligibility Criteria

Only studies with human beings, in English language, evaluating the relationship between health professionals and patient safety were included. No study protocols or qualitative research were included as they do not present quantitative results. Author letters, theses, dissertations, and research on animals were excluded.

### 2.4. Statistical Analysis of Meta-Analysis

The effect size of continuous outcomes was evaluated using the weighted mean difference (WMD) and dichotomous outcomes assessed using the risk ratio (RR) with a 95% confidence interval (CI). Heterogeneity was evaluated with I^2^ statistics. A random-effects model was applied regardless of the heterogeneity of results.

The meta-analysis was performed with the Stata 14.0 program (StataCorp LLC, College Station, TX, USA) using the random-effects model. A meta-analysis of proportion and metabias was also performed for effect analysis based on the standard error.

A data quality assessment was performed, and the screening process was also executed by other researchers, as well as the statistical analysis of the data by a third statistician.

## 3. Results

The first search generated a total of 124 articles. After filtering by reading titles, abstracts, and duplicates, 95 studies were excluded and 29 articles were selected for a reading of the full text. Of these, 21 articles met the inclusion criteria, as is illustrated in Figure 1.

### Study Characteristics

The selected studies were carried out mostly with nurses and physicians, mainly in the United States, followed by European countries; 90% were cross-sectional studies, 5% were prospective, and 5% longitudinal, with studies between the years 2006 and 2018 (Table 1).

Table 2 illustrates the purpose of the selected studies that met the inclusion criteria and their main findings.

The analysis demonstrated a relationship between burnout and patient safety actions. There was an effect of 2.67 with a confidence interval of 2.3 to 3.0, which represents a probability of superiority of 96.7% (Figure 2).

The analysis of proportions demonstrated an effect on patient safety actions of 0.60, evidencing a probability of superiority of 66.4%. It represents an average effect based on effect interpretation and associates the development of burnout in health professionals with patient safety actions (Table 3).

Most of the studies were within an acceptable standard error in health with representativeness in the range of 95%, reinforcing the congruence with the meta-analysis and assertiveness in the general interpretation of results (Figure 3).

## 4. Discussion

The results revealed an association between burnout and patient safety permeating the work process, personal characteristics, and teamwork [20,21,22,23,24,25,26,27,28,29,30,31,32,33,34,35,36,37,38,39,40]. Although some studies indicate this association in a less significant way, the meta-analysis shows an association of more than 60%.

Studies corroborate that the presence of burnout leads to a decline in patient safety [21,24,28,30,31,35,36,37]. In units with higher burnout scores, there was a deterioration of teamwork climate, safety, and job satisfaction [40]. Professional tiredness, one of the exhaustion criteria, implies a lower ability for effective teamwork, which negatively affects patient safety [28].

Higher levels of burnout were also associated with unfavorable outcomes, patient dissatisfaction, and increased patient and family complaints [31]. This can be explained by emotional fatigue and depersonalization, which trigger in the health professional the feeling of exhaustion and cynicism, becoming distant and cold in front of the patients’ needs, which compromises the quality of care.

Although most studies confirm this association, Garrouste-Orgeas et al. [29] found no influence of burnout on the occurrence of medical errors, nor did they identify an association between the disease and patient safety culture scores. Tawfik et al. [22] observed that general healthcare associated infections (HAI) rates were not associated with burnout, but the feeling of working hard.

Rodrigues, Santos, and Sousa [41] specified that the work environment as a source of stress, excessive workload, and lack of organizational support are the main contributing factors for this association. This was confirmed by Holden et al. [34] who pointed out that external efforts negatively impacted burnout and patient safety, but personal or internal efforts were not associated with the occurrence of adverse events and exhaustion.

The workload is indicated as a determining factor for professional fatigue. Specifically, for health professionals with high burnout rates, high hourly loads had a strong negative impact on patient safety [33,35].

This information can be evidenced when trying to understand the elements that predicted hospital infections, since when directing the research instrument to nurses, it was identified that a third of them had burnout, and this variable was associated with urinary tract infection; if a nurse has burnout, each additional patient included in their workload will be exposed to a higher risk of infection [42].

Among the factors identified by the professionals, the problems related to changes in their respective health services, the imbalance between personal and professional life, and their consequences for the quality of care and patient safety stand out. More specifically, physicians identify that work pressure and hierarchy are detrimental aspects, while nurses are primarily concerned with their quality of life and patient safety at work [24,37,40].

Carayon et al. [43] pointed out 58 risk categories that involve working conditions in a hospital environment and the influence of human and systemic factors on the occurrence of errors, where some of them are lack of professionalism, high workload, poor use of tools and technologies, messy work space, and the hierarchical culture, which in many cases still prevails, contributing to a disjointed service; these circumstances were also evidenced in other results [28].

On the other hand, on nurses’ complaints related to the hierarchy of medical level, Oliveira et al. [44] noticed that the main generators of the problems of personnel conflict, besides doctors, are the nurses, managers, or professionals who have more years of experience in the ward or have a leadership position.

In this line, teamwork is evidenced as one of the important points that prevent the development of burnout in professionals, and consequently, behaviors that may harm the patient’s safety. However, the disarticulation of teamwork can also lead to the opposite, as Welp, Meier, and Mansen [28] showed that emotional and professional exhaustion has an undesirable effect on teamwork.

It can be observed that one of the major issues for the development of burnout focuses on the organizational flow of work that consequently influences personal satisfaction, patient safety culture, and psychological aspects such as mental exhaustion.

A positive safety culture toward the patient was associated with the absence of burnout and a high capacity to deal with stressful situations [25]. Open communication, management support, professional suitability, mutual learning, and teamwork are considered dimensions of safety culture; it is likely that the professional inserted in this scenario feels productive and satisfied, and therefore exposed to a lower chance to develop burnout.

In view of this, the use of leadership walk rounds is a routine that aims to periodically establish a relationship between leaders (heads of health services) and professionals who work in patient care, and to identify and solve problems involving the care and patient safety. Sexton et al. [20,32] confirm in different dimensions and moments the positive influence that this routine exerts on the negative perception of burnout, improving patient safety results.

Another point of conflict, especially for nurses, is autonomy and leadership at work. Spence and Leiter [38] see nursing leadership as a positive point to effectively affect patient safety outcomes, in addition to having good interpersonal relationships, reduces the chances of exhaustion and prevents professional fatigue.

In view of the above, it is possible to highlight the emerging need for strategies aimed at controlling burnout and especially its association with patient safety. The protocol for training implementation called TeamSTEPPS (Team Strategies and Tools to Improve Patient Safety and Performance) aims to improve the psychological safety of the team, reduce burnout among professionals, and extend the report of errors [39].

These results highlight the importance of working under the precepts of health promotion, as this will help with positively affecting the patient’s safety and the quality of life of professionals and the population [45,46,47,48].

This study, as well as routines of improvement of organizational workflow in health, strengthens the need for interventions for the improvement of health services given the large number of burnout cases observed and their direct association with patient safety. The development of new studies aimed at identifying this association more closely in all health professionals, regardless of their line of work, is extremely relevant for health interventions.

## 5. Conclusions

The presence of burnout among health professionals is associated with worsening patient safety. High levels of burnout is related to external factors, such as high workload, long hours, and interpersonal relationship.

Avoiding professional exhaustion is an important strategy for improving patient safety.

## Figures and Tables

**Figure 1 medicina-55-00553-f001:**
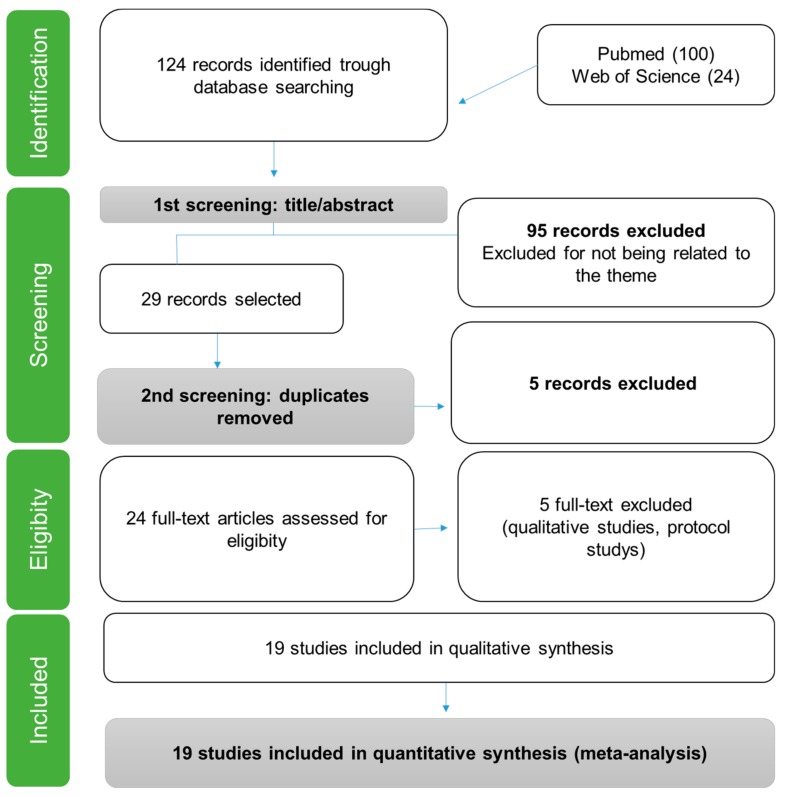
Flow diagram of literature review.

**Figure 2 medicina-55-00553-f002:**
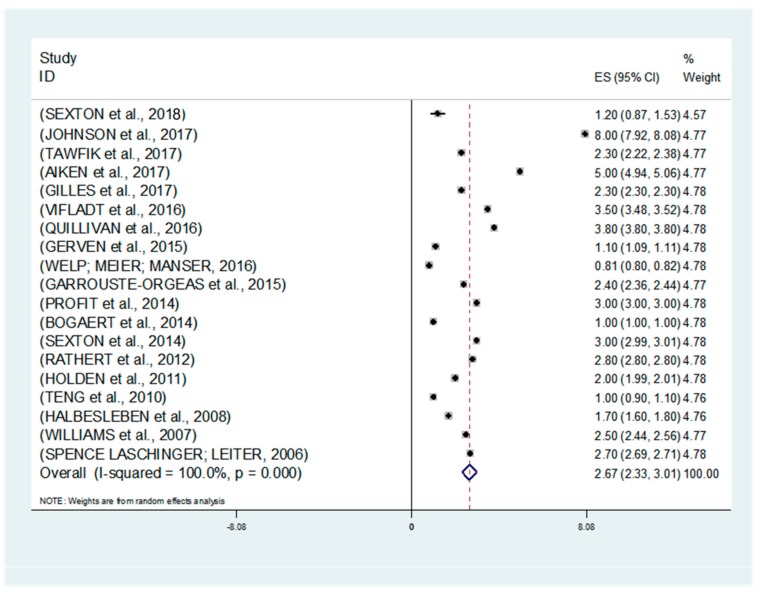
Meta-analysis: random effects.

**Figure 3 medicina-55-00553-f003:**
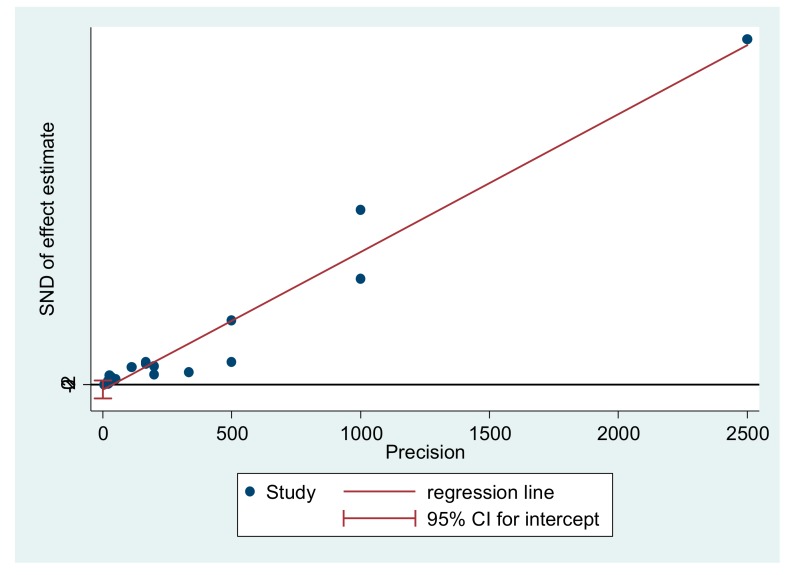
Metabias. SDN: Standard deviation.

**Table 1 medicina-55-00553-t001:** Design of selected studies and location.

Authors	Year	Country/Location	Method
Sexton et al. [20]	2018	United States	Cross-sectional study
Johnson et al. [21]	2017	United Kingdom	Cross-sectional study
Tawfik et al. [22]	2017	United States	Cross-sectional study
Aiken et al. [23]	2017	Belgium, England, Finland, Ireland, Spain, and Switzerland	Cross-sectional study
Gilles, Courvoisier, and Peytremann-Bridevaux [24]	2017	Switzerland	Cross-sectional study
Vifladt et al. [25]	2016	Norway	Cross-sectional study
Quillivan et al. [26]	2016	United States	Cross-sectional study
Gerven et al. [27]	2016	Belgium	Cross-sectional study
Welp, Meier, and Manser [28]	2016	Switzerland	Longitudinal study
Garrouste-Orgeas et al. [29]	2015	France	Prospective, observational, and multicentric Study
Profit et al. [30]	2014	United States	Cross-sectional study
Bogaert et al. [31]	2014	Belgium	Cross-sectional study
Sexton et al. [32]	2014	United States	Cross-sectional study
Rathert et al. [33]	2012	United States	Cross-sectional study
Holden et al. [34]	2011	United States	Cross-sectional study
Teng et al. [35]	2010	Taiwan	Cross-sectional study
Halbesleben et al. [36]	2008	United States	Cross-sectional study
Williams et al. [37]	2007	United States	Cross-sectional study
Spence Laschinger and Leiter [38]	2006	Canada	Cross-sectional study

**Table 2 medicina-55-00553-t002:** Studies selected, objectives and main results.

Authors	Objective	Main Results
Sexton et al. [20]	Evaluate the associations between receiving comments on actions taken as a result of the patient safety visit routine walk rounds (WR) and health worker assessments of patient safety culture, employee involvement, burnout, and work–life balance life.	Feeling as if health professionals have minimal control over quality of care through patient safety visit routine walk rounds (WR) can reduce their own perceptions of burnout, for example, that they are working too hard or feeling frustrated at work. These findings indicate that feedback WR may provide a significant opportunity to reduce fatigue.
Johnson et al. [21]	To investigate the relationships between depressive symptoms, burnout, and patient safety perceptions.	When tested in separate analyses, depressive symptoms and facets of burnout were associated with patient safety measures. In addition, the proposed mediation model was supported, with associations between depressive symptoms and patient safety perceptions fully mediated by burnout.
Tawfik et al. [22]	To examine the prevalence of burnout among California’s neonatal intensive care units (NICUs) and test the relationship between infection due to burnout and healthcare associated infections (HAI) rates in very low birth weight infants (VLBW).	Variable prevalence of burnout was found in the surveyed NICUs (mean 25.2 ± 10.1%). HAI rates were 8.3 ± 5.1% during the study period. The highest prevalence of burnout was found among nursing professionals and respiratory therapists, day shift workers, and workers with five or more years of service.
Aiken et al. [23]	To determine the association of hospital nursing skill mixed with patient mortality, care patient assessments, and quality of care indicators.	In a standard hospital, almost 30% of nurses scored high on the burnout scale, and a similar percentage expressed dissatisfaction with their jobs.
Gilles, Courvoisier, and Peytremann-Bridevaux [24]	To qualitatively analyze the open comments included in a job satisfaction survey and align these with quantitative results.	About a third of the comments addressed scheduling issues, mainly related to change-related problems and exhaustion, work–life balance, difficulties with colleagues’ absences, and the consequences for quality of care and patient safety. While some comments were provided equally by all professional groups, others were group-specific, as follows: work pressures and hierarchy observed by physicians, quality of health and patient safety noted by nurses, and skill recognition mentioned by administrative staff.
Vifladt et al. [25]	Examine the relationship between the perception of registered nurses (RNs) on patient safety culture, burnout, and sense of coherence, and compare burnout and the sense of coherence in ICUs restructured and not restructured.	A positive safety culture was statistically significantly associated with a low burnout score and a strong sense of coherence. No statistically significant differences were found in burnout and sense of coherence between restructured and non-restructured ICUs.
Quillivan et al. [26]	To evaluate the influence of patient safety culture on the distress related to the second victim.	Of the 358 nurses from a specialized pediatric hospital, 169 (47.2%) completed two surveys (patient safety culture and the second victim experience and support tool). Hierarchical linear regression demonstrated that the size of the patient safety culture survey and non-positive response to error was significantly associated with reductions in the second victim’s psychological, physical, and occupational survey dimensions (*p* < 0.001). As a mediator, organizational support fully explained the nonpunitive response to error, physical anguish and the nonpunitive response to professional-error relationships, and partially explained the unpalpable response to the psychological-error relationship.
Gerven et al. [27]	To investigate the prevalence of health professionals personally involved in a patient safety incident (PSI), as well as the relationship of involvement and degree of harm with problematic medication use, excessive alcohol consumption, risk of burnout, work–home interference (WHI), and turnover intentions	Nine percent of the total sample was involved in an PSI during the previous 6 months. Involvement in a PSI was related to a higher risk of burnout (β = 0.40, OR = 2.07), to problematic drug use (β = 0.33, OR = 1.84), to higher WHI (β = 0.24), and higher turnover intentions (β = 0.22). Injury to the patient was a predictor of problematic medication use (β = 0.14, OR = 1.56), risk of burnout (β = 0.16, OR = 1.62), and WHI (β = 0.19).
Welp, Meier, and Manser [28]	The study focused on the long-term development of teamwork, emotional exhaustion, and patient safety in interprofessional intensive care teams, exploring the causal relationships between these constructs. A secondary objective was to break down the effects of interpersonal and cognitive behavioral teamwork.	Emotional exhaustion had a lagged effect on interpersonal teamwork. In addition, interpersonal and cognitive behavioral teamwork influenced each other. Finally, cognitive behavioral teamwork predicted clinician-rated patient safety.
Garrouste-Orgeas et al. [29]	To assess whether burnout, depression symptoms, and safety culture affect the frequency of medical errors and adverse events in ICUs.	The symptoms of depression were an independent risk factor for medical errors. Burnout was not associated with medical errors. The safety culture score had a limited influence on medical errors. Other independent risk factors for medical errors or adverse events were related to ICU organization (40% of ICU staff out of work the day before), staff (safety-specific training), and patients (workload).
Profit et al. [30]	Examine the relationships between burnout in neonatal intensive care units (NICUs) and patient safety culture.	The percentage of participants in each NICU reporting burnout ranged from 7.5% to 54.4%. Burnout varied significantly between NICUs, *p* < 0.0001, but was less prevalent in physicians compared to non-physicians. NICUs with higher burnout scores had a lower teamwork climate, safety climate, job satisfaction, management perceptions, and working conditions.
Bogaert et al. [31]	Investigate the impact of factors of the nurse’s practice environment, nurse work characteristics, and nurse burnout reported work outcomes, quality of care, and patient adverse event variables at the nursing unit level.	Several unit-level associations (simple models) were identified between factors of the nurse’s practice environment, nurse’s work characteristics, burnout dimensions, and outcome variables reported by the nurse. Multiple multilevel models showed several independent variables, such as unit-level nursing management, social capital, emotional exhaustion, and depersonalization, as important predictors of nurse-reported outcome variables, such as job satisfaction, turnover intentions, quality, patient and family complaints, patient and family verbal abuse, patient falls, nosocomial infections, and medication errors.
Sexton et al. [32]	Compare stress perceptions and intensity among hospital shift nurses across three countries: Israel, USA (Ohio State), and Thailand.	The patient safety leadership walk rounds tool has been associated with improved safety culture outcomes and lower NICU burnout rates.
Rathert et al. [33]	Based on the resource conservation theory, the study examined a conceptual model that links the job environment with alternative solutions in acute care nurses and other clinicians, and the hypothesis that burnout (specifically emotional exhaustion) intervene this relationship.	The hypotheses were examined using structural equation modeling. Time pressure was positively related to exhaustion, and autonomy was negatively related. Exhaustion was positively related to alternative solutions and mediated the time pressure and autonomy for alternative solution relations. Contrary to expectations, the physical environment was directly and negatively related to alternative solutions.
Holden et al. [34]	This study sought to measure the effect of workload on safety and outcomes of workers in two pediatric hospitals using a new approach to workload measurement.	Pharmacists and pharmacy technicians reported high levels of external and internal mental demands during dispensation. The study supported the hypothesis that external demands (interruptions, divided attention, and running) negatively impacted medication safety and employee welfare outcomes. However, as hypothesized, increasing levels of internal demands (concentration and effort) were not associated with a higher perceived probability of error, adverse drug events, or burnout, and even had a positive effect on job satisfaction.
Teng et al. [35]	Investigate how time pressure and the interaction of nurse’s time pressure and burnout affect patient safety.	While regression analysis results suggest that time pressure did not significantly affect patient safety (β = −0.0, *p* > 0.05), time pressure and burnout had an interactive effect on patient safety (β = −0.08, *p* < 0.05). Specifically, for nurses with high burnout (n = 223), time pressure was negatively related to patient safety (β = −10, *p* < 0.05).
Halbesleben et al. [36]	To analyze the relationship between burnout and patient safety indicators in nurses.	After controlling for work-related demographics, multiple regression analysis supported the prediction that burnout was associated with perceived lower patient safety. Burnout was not associated with event reporting behavior but was negatively associated with reporting errors that did not lead to adverse events.
Williams et al. [37]	To investigate the cultural conditions that affect medical stress, dissatisfaction, and burnout syndrome by examining whether they offer poor quality of care.	Cultural emphasis on quality played a key role in quality outcomes. In addition, it was found that stressed and dissatisfied doctors report a higher probability of making mistakes and more frequent cases of sub-optimal patient care, associating this result with burnout syndrome.
Spence Laschinger and Leiter [38]	To test a theoretical model of professional nursing environments linking the conditions of professional nursing practice to burnout, and subsequently, to patient safety outcomes.	Nursing leadership played a key role in the quality of working life in relation to political involvement, staffing levels, support for a nursing care model (vs. physician), and nurse/physician relationships. Staff adequacy directly affected emotional exhaustion, and the use of a nursing care model had a direct effect on nurses’ personal fulfillment. Both directly affected patient safety outcomes.

**Table 3 medicina-55-00553-t003:** Meta-analysis of proportions.

Study	ES	95% Conf. Interval		%Weight
Sexton et al. [20]	0.32	0.32	0.33	4.84
Johnson et al. [21]	0.77	0.73	0.82	4.78
Tawfik et al. [22]	0.72	0.70	0.74	4.83
Aiken et al. [23]	0.77	0.76	0.78	4.84
Gilles. Courvoisier. and Peytremann-Bridevaux [24]	0.42	0.41	0.44	4.84
Vifladt et al. [25]	0.49	0.44	0.55	4.74
Quillivan et al. [26]	0.47	0.42	0.52	4.76
Gerven et al. [27]	0.52	0.51	0.53	4.84
Welp. Meier. and Manser [28]	0.71	0.69	0.73	4.83
Garrouste-Orgeas et al. [29]	0.52	0.50	0.55	4.82
Profit et al. [30]	0.51	0.49	0.53	4.83
Bogaert et al. [31]	0.72	0.69	0.75	4.82
Sexton et al. [32]	0.72	0.70	0.74	4.83
Rathert et al. [33]	0.50	0.47	0.53	4.81
Holden et al. [34]	0.73	0.67	0.78	4.73
Teng et al. [35]	0.49	0.44	0.53	4.77
Halbesleben et al. [36]	0.50	0.40	0.60	4.55
Williams et al. [37]	0.59	0.54	0.63	4.77
Spence Laschinger and Leiter [38]	0.70	0.69	0.71	4.84
Random pooled ES	0.60	0.51	0.68	100.0

Heterogeneity chi^2 = 9815.67 (d.f. = 20) *p* = 0.000. I^2 (variation in ES attributable to heterogeneity) = 9980%. Estimate of between study variance Tau^2 = 0.04 Test of ES = 0; z = 13.73; *p* = 0.00.

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
