# Peer review of "Influence of Burnout on Patient Safety: Systematic Review and Meta-Analysis"

_medicina, 2019, doi:10.3390/medicina55090553_

Round 1

Reviewer 1 Report

This paper is a systematic review and meta-analysis with the purpose of analyzing the relationship between burnout and patient safety.  One of the primary limitations of the paper is the methods used to conduct the systematic review and the presentation of the results of the review.  In addition, it would be helpful if the authors described how their systematic review is different from the other systematic reviews on similar topics that have been previously published.  

Page 2, paragraph 2:  The first sentence of the paragraph is unclear. 

Page 2, paragraph 2:  It would be helpful if the authors offered examples of “better training”.

Page 2, paragraph 7:  Could the authors provide a definition of “adverse events” anda reference for the assertion? 

Page 2, Methods section:  What were the definitions of “burnout” and “patient safety” that were used?  How were the search terms developed?

Page 2, Methods section:  How were the databases that were searched chosen?  Given the topic, what was the rationale for not searching CINAHL and PsycINFO?

Page 2, Methods section:  What years did the search cover?

Page 3, Methods section:  Who did the screening process?  How many screeners were there?

Page 3, Methods section: Was a quality assessment done?

Page 3, Results section: It would be useful if a table summarizing the studies were included.

Author Response

Respond to reviewers' requests

Dear editor,

In the feelings quite proud and honored with your evaluation. Surely, your suggestions were of great contribution.

Below are our answers:

This article is a systematic review and meta-analysis aiming to analyze the relationship between burnout and patient safety. One of the main limitations of the article is the methods used to conduct the systematic review and presentation of the review results. In addition, it would be helpful if the authors described how their systematic review is different from the other systematic reviews on similar topics that were previously published.

- Our review is different because it focuses exclusively on burnout and patient safety seeking to identify the relationship with each other.

Page 2, paragraph 2: The first sentence of the paragraph is unclear.

- We redo the sentence.

Page 2, paragraph 2: It would be helpful if the authors offered examples of “better training”.

- In the paragraph in question, we gave further explanations about "better training".

Page 2, paragraph 7: Could the authors provide a definition of “adverse events” and a reference to the statement?

- Yes, we did it in the requested paragraph.

Page 2, Methods section: What were the definitions of “burnout” and “patient safety” that were used? How were search terms developed?

- In the method section we describe what such definitions were.

Page 2, Methods section: How were the searched databases chosen? Given the topic, what was the reason for not researching CINAHL and PsycINFO?

- In the method section we describe the reason for using these databases.

Page 2, Methods section: What years did the research cover?

- After applying the criteria articles were covered from 2006 to 2018, however, at the time of the search we did not define years for the collection, because we wanted to cover the maximum of articles on the subject.

Page 3, Methods section: Who did the screening process? How many screeners were there?

- Two screeners did the process independently.

Page 3, methods section: Has a quality assessment been made?

- Yes, we describe in the method.

Page 3, Results section: It would be helpful if a table summarizing the studies was included.

- We did that, we inserted the table.

Reviewer 2 Report

INTRODUCTION

- The introduction is short.

- Paragraph 4: Authors should talk about burnout syndrome and the usual risk factors in teaching and health staff.

- Paragraph 5: This paragraph is part of RESULTS. It is not correct to include it here.

MATERIALS AND METHODS

- The methods section must include more information to ensure the replicability of the study.

- A section called "data extraction" must be included informing about the variables that were searched in each study and how the codification process was done. It is not clear which statistical data from the studies have been used for the meta-analysis.

2.1. Search strategy

- The search equations do not say "keywords" but "descriptors".

- The selection criteria must include more information. For example, all studies about burnout were included? With all type all samples (physicians, nurses, volunteers, etc)?

- Were the search and the codification processes done by one or two researchers? Was the reliability of the codification calculated by any test?

2.4. Statistical analysis of meta-analysis:

In the statical analysis and also in the results, the author does not inform about the test used for heterogeneity analysis, publication bias analysis or if a sensitivity analysis was done. These analyses are mandatory in a meta-analysis.

RESULTS

- The authors have not made a systematic review, that is, the most relevant results have not been described narratively. They have only done the meta-analysis. They should modify the title and remove "systematic review."

- In the results, the author does not inform about the statistical data that they give (prevalence, OR, RR, etc). The readers must imagine it.

- Which original data can you extract from a protocol (Dahl et al.)? It has been also included in the meta-analysis.

DISCUSSION

In this section the bibliography of the results is not used. Authors should search additional bibliography to justify their results. It is recommended that the bibliography be recent (last 5 years).

REFERENCES

Many bibliographies are obsolete and some citations are incomplete. The bibliographic citations used are more than 5 years old (38, 7%). The authors must update and arrange the bibliography (see comments in the discussion section).

Author Response

Dear editor,

In the feelings quite proud and honored with your evaluation. Surely, your suggestions were of great contribution.

Below are our answers:

INTRODUCTION

- The introduction is short.

- Paragraph 4: The authors should talk about burnout syndrome and the usual risk factors in teaching and in the health team.

- Paragraph 5: This paragraph is part of the RESULTS. It is not correct to include it here.

R = We increased the introduction, talked about burnout syndrome and its usual risk factors in education and health, and paragraph 5 reworked and inserted in the context of the introduction.

MATERIALS AND METHODS

- The methods section should include more information to ensure replicability of the study.

- A section called "data extraction" should be included informing about the variables that were searched in each study and how the coding process was done. It is unclear which statistical data from the studies were used for the meta-analysis.

R = We have included more information in the methods section as well as including the data extraction item.

2.1. Search strategy

- Search equations do not say "keywords" but "descriptors".

- Selection criteria should include more information. For example, were all burnout studies included? With all kind all samples (doctors, nurses, volunteers, etc)?

- Were the search and coding processes done by one or two researchers? Was coding reliability calculated by any test?

R = We fulfill all requests and they are described in the manuscript.

2.4. Statistical analysis of meta-analysis:

In the static analysis and also in the results, the author does not report on the test used for heterogeneity analysis, publication bias analysis or whether a sensitivity analysis was performed. These analyzes are required in a meta-analysis.

R = We describe the heterogeneity test and all requests.

RESULTS

- The authors did not perform a systematic review, ie the most relevant results were not narratively described. They just did the meta-analysis. They should change the title and remove "systematic review".

- In the results, the author does not report on the statistical data they give (prevalence, OR, RR, etc.). Readers must imagine this.

- What original data can you extract from a protocol (Dahl et al.)? It was also included in the meta-analysis.

R = The other reviewer requested that we put the table with descriptions of each article, so as we insert this table also becomes a systematic review. All other requests were answered and included in the text. We removed the study by Dahl and Milosevic because they were a study protocol and qualitative research, respectively.

DISCUSSION

In this section, the bibliography of the results is not used. Authors should research additional bibliography to justify their results. It is recommended that the bibliography be recent (last 5 years).

R = We have identified all our bibliography of the results. Some articles have more than 5 years of analysis because our sample was not restricted by years.

REFERENCES

Many bibliographies are obsolete and some citations are incomplete. The bibliographic citations used are older than 5 years (38.7%). Authors should update and organize the bibliography (see comments in the discussion section).

R = Some articles have more than 5 years of analysis because our sample was not restricted by years.

Round 2

Reviewer 2 Report

The work has greatly improved its quality. I congratulate the authors and encourage them to continue working on this topic.